# The Relationship of Lake Morphometry and Phosphorus Dynamics of a Tropical Highland Lake: Lake Tana, Ethiopia

**Mebrahtom G. Kebedew** [1,2]**, Aron A. Kibret** [1,3,4]**, Seifu A. Tilahun** [1]**, Mulugeta A. Belete** [1]**, Fasikaw A. Zimale** [1] **and Tammo S. Steenhuis** [1,5,*]

[1] Faculty of Civil and Water Resources Engineering, Bahir Dar Institute of Technology, Bahir Dar University, Bahir Dar 6000, Ethiopia; mebrehydro@gmail.com (M.G.K.); atekaaron@gmail.com (A.A.K.); satadm86@gmail.com (S.A.T.); mulugetaazeze94@gmail.com (M.A.B.); fasikaw@gmail.com (F.A.Z.)

[2] School of Civil Engineering, Ethiopian Institute of Technology, Mekelle University, Mekelle 7000, Ethiopia

[3] Faculty of Chemical and Food Engineering Bahir Dar Institute of Technology, Bahir Dar University, Bahir Dar 6000, Ethiopia

[4] Amhara Bureau of Water Resources, Irrigation and Energy, Bahir Dar 6000, Ethiopia

[5] Department of Biological and Environmental Engineering, Cornell University, Ithaca, NY 14850, USA

[*] Correspondence: tss1@cornell.edu; Tel.: +1-607-255-2489

**Abstract:** Lakes hold most of the world's fresh surface water resources. Safeguarding these resources from water quality degradation requires knowledge of the relationship between lake morphometry and water quality. The 3046-km$^2$ Lake Tana in Ethiopia is one of the water resources in which the water quality is decreasing and water hyacinths have invaded. The objective of this study is to understand the interaction between the lake morphometry and water quality and specifically the phosphorus dynamics and their effect on the water hyacinths. A bathymetric survey was conducted in late 2017. Various morphometric parameters were derived, and both these parameters and sediment available phosphorus were regressed with the dissolved phosphorus. The results show that, with a wave base depth that is nearly equal to a maximum depth of 14.8 m, the bottom sediments were continuously suspended in the water column. As a result of the resuspension mixing, we found that the dissolved phosphorus in the water column decreased with lake depth and increased with sediment available phosphorus ($R^2 = 0.84$) in the northern half of the lake. This relationship is not as strong in the south due to a large flow of Gilgel Abay to the outlets. Water hyacinths were found where the lake was shallow and the available phosphorus was elevated. The large reservoir of sediment phosphorus will hamper any remedial efforts in removing the water hyacinths.

**Keywords:** Lake Tana; bathymetry; morphometry; phosphorus; water hyacinth; tropical lake

## 1. Introduction

Lakes are important ecosystems whose environments are continuously changing due to natural and human factors [1,2]. The rate of change depends in part on the morphometric characteristics [1,3,4]. For this reason, Lake Tanganyika and Lake Malawi, which are deep, have been less affected by the degradation of the watershed and increased fertilizer use [5,6] than shallow lakes such as Lake Chad and Lake Tana [7,8]. The difference is that deep lakes are stratified [9,10] and shallow lakes are mixed by the wind and waves [2,11]. Another morphometric characteristic that affects water quality is the shape of the lake. Irregular shaped lakes have longer shoreline lengths than circular lakes and therefore are more vulnerable to the anthropogenic impacts of shore development [12]. Many scholars have developed empirical models for defining lake morphometry in relation to water quality [1,4,12–14].

Eutrophication of lakes is caused by nutrient enrichment and principally dissolved phosphorus. The dissolved phosphorus concentrations are directly related to morphology and, specifically, the wave base depth, which is the maximum depth at which a water wave's passage causes significant water motion. At water depths deeper than the wave base, bottom sediments are no longer stirred by the wave motion above [10,15,16]. In cases in which the wave base depth is greater than the lake depth, the bottom sediment is resuspended in the water column [17,18] and the dissolved phosphorus is affected by the available phosphorus in the bottom sediment [14,19]. Therefore, large shallow lakes are prone to sediment resuspension and phosphorus release from the sediment [9–11,20].

One of the lakes in which the water quality is rapidly declining is the 3046 km$^2$ Lake Tana [8]. Sediment deposition and phosphorus concentrations have been increasing over the last 50 years [21,22]. The lake is in a transitional stage to eutrophic status [23] and water hyacinths have appeared [24]. Previous studies have addressed the lake water balance [25–27], the total sediment mobilized from the upland watersheds [28,29], sediment budgets [30,31], bottom sediment characteristics [32], peninsula development [33,34], dissolved and available phosphorus concentrations in the lake water and bottom sediments [32,35], water quality and status of eutrophication condition [8,23] and acreages of water hyacinths [24,36,37]. However, the relationship between the lake's physical characteristics and water quality has not yet been investigated. The general objective of this research is to determine Lake Tana's morphometric parameters and their implication on water quality and specifically to investigate sediment and phosphorus dynamics and their implications on eutrophication and the spread of water hyacinths on Lake Tana by correlating the total dissolved phosphorus concentration in the lake water as a function of lake depth and available phosphorus in the bottom sediments.

To do so, a bathymetric survey was conducted in late 2017, and dissolved phosphorus concentrations were related to morphometric parameters and available phosphorus in the bottom sediment. The study will aid further in the understanding of the spread of water hyacinths in tropical lakes.

## 2. Materials and Methods

### 2.1. Study Area

Lake Tana is located in the Northwestern Ethiopian Highlands. It is the largest freshwater lake in Ethiopia and the third largest in the Nile Basin [29]. The catchment area of Lake Tana is 12 thousand square kilometer. The water level is the highest in September and decreases by around 3 m with the lowest level in May [38]. The lake covers 20% of the total catchment.

The lake is a natural reservoir for several hydropower plants: the Tis Abay I and II hydropower plants are located near the Blue Nile falls, 30 km south of the lake and the 460 kW Tana Beles plant, located west of the lake, which became operational in 2010 [38,39]. The Blue Nile, starting in 1995, has been regulated by Chara Chara Weir. Water for the Tana Beles plant flows through a 26 km tunnel to the Beles River.

The lake has low water transparency due to the high sediment load of the inflowing rivers during the rainy phase (June-September) and the re-suspension of sediment [29]. The gross primary production rate of Lake Tana is among the lowest of the tropical lakes [40]. The lake water is slightly basic, with an average pH of 8.2 and average annual temperature of 23 °C [36]. Phosphorus flux in the main rivers draining to Lake Tana is increasing [22] and the lake has an average of 0.2 mg P/L of dissolved phosphorus concentration [35]. In 2011, water hyacinths, *Eichhornia crassipes*, appeared [24,41]. The spatial distribution of the water hyacinths in Lake Tana is dynamic with lake water levels [42].

According to Wosenie et al. [25], the annual water balance of Lake Tana consists of 6.8 km$^3$ inflow, 5.0 km$^3$ outflow, 4.1 km$^3$ from precipitation and the evaporation is 5.5 km$^3$ from the lake. Mamo et al. [43] suggests that nearly 1 km$^3$ is lost through faults at the western edge of Lake Tana and, to a lesser degree, through the fault near the Blue Nile outlet. Based on the inflow data, over 23% of the lake volume is exchanged each year. The Gilgel Abay is the largest river entering the lake in the

southwest [25]. The Gumara and Rib Rivers, which together are nearly the same size as Gilgel Abay, enter the lake in the east and Megech, which is the smallest of the four rivers, in the north. [29].

### 2.2. Datasets

A bathymetric survey was conducted to obtain the data for morphometric analysis. Spatially distributed phosphorus concentrations for the lake were published previously by the authors separately. The available phosphorus in sediment was documented in [32] and the dissolved phosphorus was measured by Kibret for his MSc thesis and published as [36].

### 2.2.1. Bathymetric Data

We conducted a bathymetric survey from 20 September to 3 October and from 2 to 5 November in 2017. This survey was used to calculate the morphometric characteristics of Lake Tana. The first survey was conducted for the whole lake on an approximate 5-km grid and the second survey was carried out by conducting one round trip of the whole lake close to the lake boundary to increase the surveying coverage for the nearshore areas (Figure 1b). Remote sensing images were downloaded from for extracting the lake boundary between land and water (Table 1). The images were downloaded when the lake was at the maximum, average and minimum levels (Figure S1 in the Supplementary Materials). The lake boundary and corresponding coordinates were then extracted from the map using ArcMap in ArcGIS (version 10.4.1) and merged with the bathymetric survey data. Lake bottom elevation was calculated by subtracting lake water depth from the lake surface level. The reference level was 1783.72 m a.s.l., located at Shum Abo Park on the southern shore of Lake Tana (Figure 1). The station is currently used to measure lake water level by the Abay Basin Authority, Bahir Dar, Ethiopia. In 2017, the minimum lake level was 1.9 m above the reference level and the maximum was 4 m above the reference height.

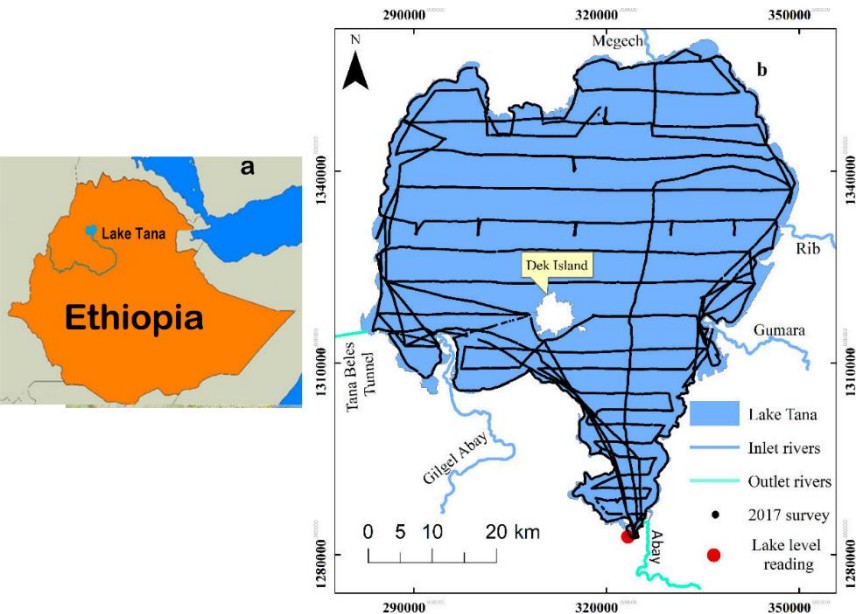

**Figure 1.** Location map of the study area: (**a**) map of Ethiopia and Lake Tana from the National Geographic world map and (**b**) Lake Tana indicating the survey routes including the four major inlets, two outlets and location of lake level reading.

**Table 1.** Description of downloaded Sentinel-2 image data for lake boundary extraction.

| Satellite | Acquisition Date | Spatial Resolution (m) | Water Level (m) | Remark |
|-----------|------------------|------------------------|-----------------|--------|
| | 29 September 2017 | 20 | 1787.71 | Water level during survey |
| Sentinel-2 | 11 February 2017 | 20 | 1786.53 | Average water level |
| | 1 June 2017 | 20 | 1785.84 | Minimum water level |

### 2.2.2. Phosphorus Concentrations

Phosphorus concentration data for the bottom sediment and lake water surface were collected from the literature (Table S1 in the Supplementary Materials). Sediment samples were taken at 60 sampling sites with an Eckman grab sampler at locations that were approximately $10 \times 10$ km grids in March 2018 [32]. Water samples for determining the dissolved phosphorus concentrations were collected from 143 sampling locations on a $5 \times 5$ km grid in March 2017.

Available phosphorus concentration in the sediment samples was determined using the Olson method [32]. Dissolved phosphorus concentration was determined by the acid-per-sulfate digestion method in the range of 0.06–3.50 mg P/L using PhosVer®3 (Hach Company, Loveland, CO, USA) [36]. The concentrations of available phosphorus in the sediment are reported on the basis of dry weight. The sampling methods and laboratory procedures are further detailed in the cited materials.

### 2.2.3. Morphometric Parameters

The morphometric parameters of Lake Tana, consisting of the maximum depth ($D_{max}$) and median depth (D50,) were derived from the bathymetric survey. The maximum length ($L_{max}$), maximum width ($B_{max}$), shoreline length ($L_o$), area (A), volume (V) and mean slope ($S_{mv}$) were measured in ArcMap. The remaining morphological parameters were derived from the measured parameters according to methods developed by Håkanson [12] (Table 2).

### 2.2.4. Interpolation

The bathymetric and phosphorus concentration data were organized in Microsoft Excel 2019 and exported to ArcMap (version 10.4.1) to determine the morphological parameters and to generate spatial maps. The 3D Analyst Tool, an extension of ArcMap, was used to determine the area and volume of the lake and exported to Excel. The elevation volume (area and elevation) was plotted in Excel. In addition, the two-dimensional (2D) contour maps and three-dimensional (3D) surface grid maps were generated in Golden surfer gridding software (version 15). Cross-validation was used as an objective function to assess the quality of interpolation techniques both in ArcMap and Surfer models. The semi-variogram model with the smallest root mean square error was selected for interpolation.

### 2.2.5. Regression Analysis

Microsoft Excel 2019 was used to calculate the descriptive statistics and regression analysis among lake depth, available sediment phosphorus and dissolved phosphorus concentrations. As the number and locations of sampling stations for the available and dissolved phosphorus were not the same, we interpolated the values to a grid of 1 km east-west and 5 km north-south (Figure S2 in the Supplementary Materials).

### 2.3. Methods

The relationship between lake morphometry and phosphorus dynamics was investigated by using ArcMap and Surfer gridding software and by regression analysis.

**Table 2.** Lake Tana morphometric parameters at the average lake level (1786.5 m a.s.l). Symbols and names are given according to Håkanson [12].

| Parameter | Symbol | Definition | Description | Values for Lake Tana | Units |
|---|---|---|---|---|---|
| Shore length | $L_o$ | | The perimeter of the lake | 431 | km |
| Area | A | Derived from Bathymetric survey and satellite data using ArcMap | Surface area | 3046 | km$^2$ |
| Volume | V | | Volume of water | 29.6 | km$^3$ |
| Median depth | $D_{50}$ | | The middle value of all depths | 10.5 | m |
| Maximum length | $L_{max}$ | | Connects the most remote shores | 80 | km |
| Maximum depth | $D_{max}$ | | The maximum lake depth | 14.8 | m |
| Mean depth | $D_{mv}$ | $\frac{V}{A}$ | The average depth of the lake water | 9.7 | m |
| Depth index | $D_{indx}$ | $\frac{D_{mv}}{D_{max}}$ | Measure for lake shape | 0.64 | |
| Average width | $B_{mv}$ | $\frac{A}{L_{max}}$ | The average width of the lake water | 46.7 | km |
| Mean effective fetch | $L_{ef}$ | $\sqrt{A}$ | The average distance of free water surface over which wind influences waves | 55.2 | km |
| Relative depth | $D_{rel}$ | $\frac{\sqrt{\pi}\,D_{max}}{20 \times \sqrt{A}}$ | Indicates lake stratification for $D_{rel} > 0.05$ | 0.024 | |
| Mean slope | $S_{mv}$ | | The average slope of the bed: 5% is a critical limit for mild and steep slope | 0.18 | % |
| Energy topography factor | ET | $0.25\,DR\,41^{\frac{0.061}{DR}}$ | The fraction of the lakebed area subjected to resuspension of fine sediments | 0.99 | |
| Wave base depth | $D_{wb}$ | $\frac{45.7\,\sqrt{A}}{21.4+\sqrt{A}}$ | Maximum depth at which a water wave's passage causes significant water motion | 14.8 | m |
| Theoretical retention time | RT | $\frac{V}{Q}$ | The time taken for complete exchange of the lake water | 4.3 | yr |

$L_{max}$: maximum length; $B_{max}$: maximum width; $B_{mv}$: mean width; $D_{50}$: median depth; $D_{mv}$: mean depth; $D_{rel}$: relative depth; $D_{wb}$: wave base depth; $L_o$: shoreline length; A: area; V: volume; DR: dynamic ratio; ET: energy topography factor for a percentage of the area exposed to resuspension of fine sediments; RT: lake retention time; Q: lake inflow.

## 3. Result and Discussion

### 3.1. Morphometric Characteristics

Downloaded satellite images and the bathymetric survey show that the lake is nearly cylindrical, with an axis of 80 km in the north-south direction and 65 km east-west (Figure 2b). At the average lake elevation of 1786.53 m a.s.l in 2017, the area of the lake was 3046 km$^2$, with a volume of 29.6 km$^3$ and a shore length of 431 km (Table 2). The average depth was 9.7 m, the median depth was 10.5 m and the maximum depth was 14.8 m. Since the length and the width are much larger than the depth, Lake Tana is classified as a shallow lake [2,44]. Consequently, evaporation of 1790 mm/a is a major component of the water balance and can be as large as 20% of the lake volume [25].

The 2D contour map and 3D surface map are shown in Figure 2. These maps were generated by ordinary kriging with an exponential semi-variogram model that had a root mean square error (RMSE) of 0.123 (Figure S3 in the Supplementary Materials). The maps show that the deepest part of the lake (at an elevation of 1773 m a.s.l.) is in the middle of the lake, north of Dek Island (Figures 1 and 3a). In the three-dimensional perspective view of Lake Tana created in Surfer, the deepest part is in dark blue and the shallowest parts have a reddish color (Figure 2b).

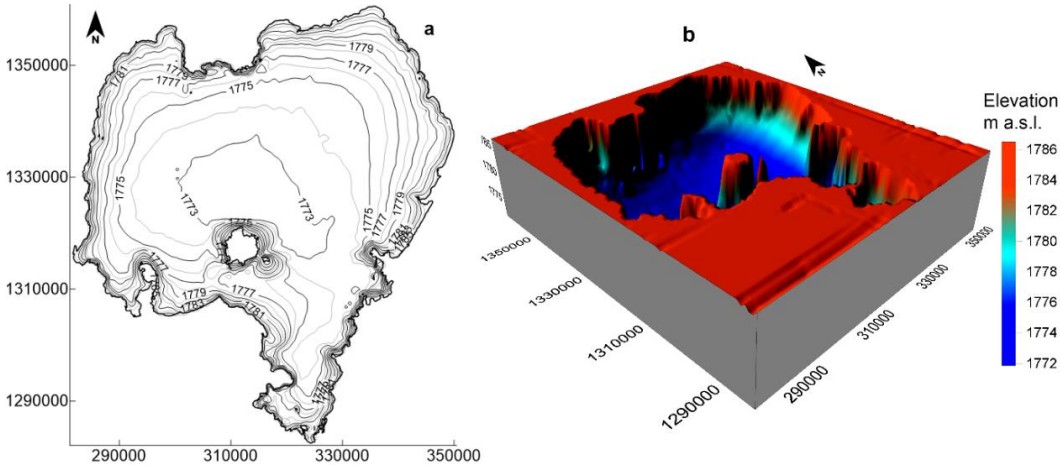

**Figure 2.** Bathymetric maps of Lake Tana (all units in maps are in meters): (**a**) two-dimensional contour maps leveled at 1 m interval; (**b**) three-dimensional colored map.

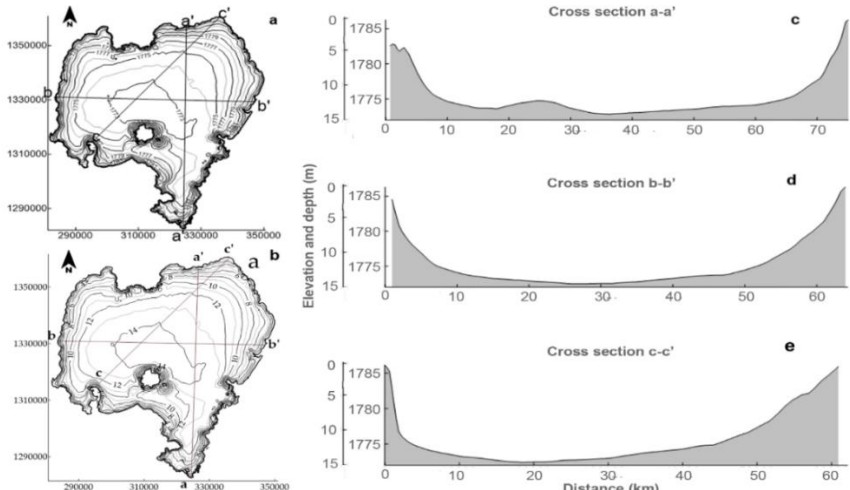

**Figure 3.** Lake elevation and depth map showing a cross-section of Lake Tana: (**a**) lake bottom contours m a.s.l.; (**b**) water depth of lake bed in below a level of 1787 m a.s.l.; (**c**) bottom profile a-a'; (**d**) bottom profile b-b'; (**e**) bottom profile c-c'.

The 2D contour map and 3D surface map are shown in Figure 2. These maps were generated by ordinary kriging with an exponential semi-variogram model that had a root mean square error (RMSE) of 0.123 (Figure S3 in the Supplementary Materials). The maps show that the deepest part of the lake (at an elevation of 1773 m a.s.l.) is in the middle of the lake, north of Dek Island (Figures 1 and 3a). In the three-dimensional perspective view of Lake Tana created in Surfer, the deepest part is in dark blue and the shallowest parts have a reddish color (Figure 2b).

The N-S, E-W and SW-NW cross-sections (Figure 3 and Figure S4 in the Supplementary Materials) show that the lakebed is relatively flat in the center and then slopes upwards near the periphery. The mean slope is 0.18% (Figure S4). Only 1% of the lake has slopes greater than 5% (Figure S4). The western side slopes are generally steeper than those of the eastern side (Figures 2 and 3c,d). The shallowest part in the northeast of the lake (Figures 2a and 3c) is invaded by water hyacinths (Figure 4c).

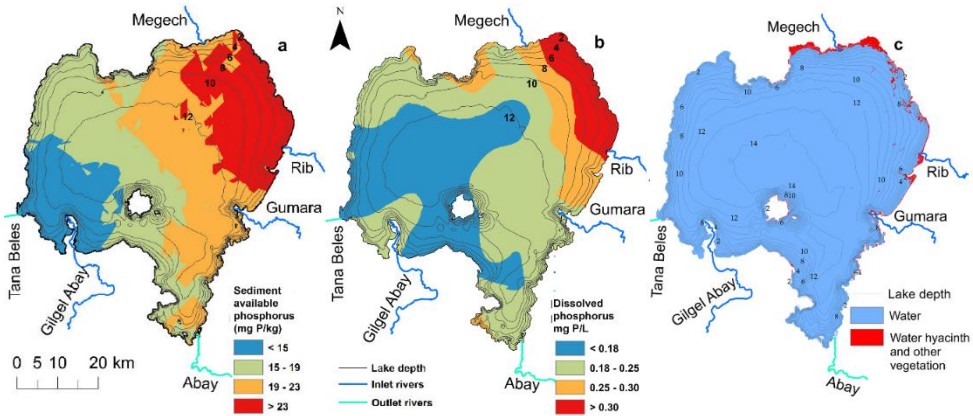

**Figure 4.** Spatial distribution of phosphorus concentration and water hyacinths of Lake Tana: (**a**) sediment available phosphorus concentration in the bottom sediments in March 2018; (**b**) dissolved phosphorus concentration in the lake water in March 2017; (**c**) water hyacinth distribution on Lake Tana in October 2019.

### 3.2. Lake Depth, Area and Volume Relationships

The relationship between the lake area and volume with elevation depicted in Figure 5 can be expressed as third-order polynomial functions (with R2 0.9972 and 0.9999, respectively):

$$A = 0.88\ (H\text{-}1772)^3 - 35.02\ (H\text{-}1772)^2 + 537.46\ (H\text{-}1772) - 62.40 \tag{1}$$

$$V = -0.005\ (H\text{-}1772)^3 + 0.21\ (H\text{-}1772)^2 + 0.088\ (H\text{-}1772) - 0.12 \tag{2}$$

where $H$ is the elevation in m a.s.l, $A$ is the area in km$^2$, and $V$ is the volume in km$^3$. About 80% of the lake area and 41% of the lake volume is located below the mean depth of 9.7 m (Figure 5, Figure S5 in the Supplementary Materials). The shallowest portion, less than 6 m deep, accounts for about 20% of the lake area. The Chara Chara Weir regulates lake level at the Blue Nile outlet so that the lake level remains between 1784 and 1787 m a.s.l, [38]. The water storage between the two levels is 8.92 km$^3$, which is about 30% of the total volume. Navigation becomes impossible when the lake level is less than 1783.75 m a.s.l [39]. In 2003, navigation ceased for four months [38].

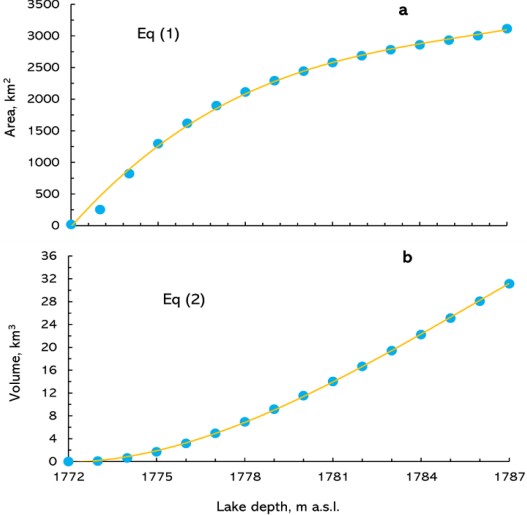

**Figure 5.** Depth–area–volume relationship of Lake Tana: (**a**) elevation–area curve; (**b**) elevation–volume curve.

### 3.3. Phosphorus Concentration in the Bottom Sediments and the Water Surface

Figure 4 shows the spatial distribution of available phosphorus in the sediment and dissolved phosphorus concentration in Lake Tana in the dry season. The average available phosphorus in the sediment was 19 mg P/kg and the dissolved phosphorus concentration in the lake water was 0.21 mg P/L in the dry season (Table S2 in the Supplementary Materials).

Figure 4a shows that the available phosphorus in the sediment is independent of the depth of the lake. For example, the most elevated concentrations of greater than 23 mg P/kg are in the shallowest part in the northeast, while the same depth at the western edge concentration is less than 15 mg P/kg. Available phosphorus concentrations of the deepest parts are between the two extremes. This distribution is linked to the morphology of the lake, the wind direction and the operation of the power plants. Since the dynamic ratio of the lake is 5.7 km/m and greater than the critical value of 3.8 km/m, the transport of sediment by resuspension takes place [12]. This means that the sediment is transported with the lake currents. The currents are stronger in the southern half than elsewhere in the lake because the water of the Gilgel Abay contributes 60% of discharge to the lake, which flows directly to the Blue Nile and Tana Beles Tunnel outlets. Thus, suspended sediment and dissolved phosphorus that came originally from the Gilgel Abay leave the lake in a relatively short time. Gumara, Rib and Megech discharge are relatively smaller and replenish the water lost by evaporation from the lake. Hence, all sediment and phosphorus originating from Gumara, Rib and Megech are deposited in the northern half. This is the reason for the high concentration of sediment phosphorus in the northern half and the low concentrations in the lake in the southern half, where the deposited sediment is resuspended and then transported away through the two outlets.

### 3.4. Relationship of Lake Depth, Available and Dissolved Phosphorus Concentrations

The distribution of the available phosphorus (Figure 4a) in March 2018 and dissolved phosphorus (Figure 4b) in March 2017 during the dry phase seem to be very similar. This is confirmed by the correlation between sediment available phosphorus and dissolved phosphorus for the northern part above Dek Island, with a coefficient of determination $R^2$ of 0.72 for a third-order polynomial (Figure 6 and Figure S2). The relationship is as follows:

$$DP = 0.0004\ (AvP)^3 - 0.02\ (AvP)^2 + 0.37\ AvP - 1.99 \tag{3}$$

where *DP* is the dissolved phosphorus concentration in mg P/L and *AvP* is the available phosphorus concentration in the sediment in mg P/kg. Equation (3) is plotted in Figure 6a and shows that the dissolved phosphorus does not change for available phosphorus concentrations between 15 and 20 mg P/kg. Only when the available phosphorus concentration is greater than 20 mg P/kg does the concentration of the dissolved phosphorus increase with increasing available phosphorus concentrations. We found a similar relationship for the lake south of Dek Island (Figure S2), but the correlation was not as strong as for the northern half, $R^2 = 0.52$ (Figure 6b), and the overall dissolved phosphorus concentrations were smaller.

Adding the depth in the regression for the northern part of Lake Tana, the coefficient of determination, $R^2$, improved to 0.84 when using a second-order polynomial for depth and available phosphorus, indicating that the depth of the lake has a significant impact on the phosphorus concentration in the water. The relationship is of the following form:

$$DP = 0.0016\ (AvP)^2 - 0.031\ AvP - 0.00056\ d^2 + 0.0080\ d + 0.46 \tag{4}$$

where *d* is depth in m. The surface of Equation (4) is shown in Figure 6c and indicates that the relationship in Equation (3) is mainly valid for depths less than 9 m. For shallow depths, the dissolved phosphorus concentrations are mainly dependent on the available phosphorus in the sediment and independent of the depth to the bottom. For greater depths, the concentration decreases both with

depth and available phosphorus. The decrease with depth is steeper when the available phosphorus is small. Adding depth to the southern part did not increase the regression coefficient.

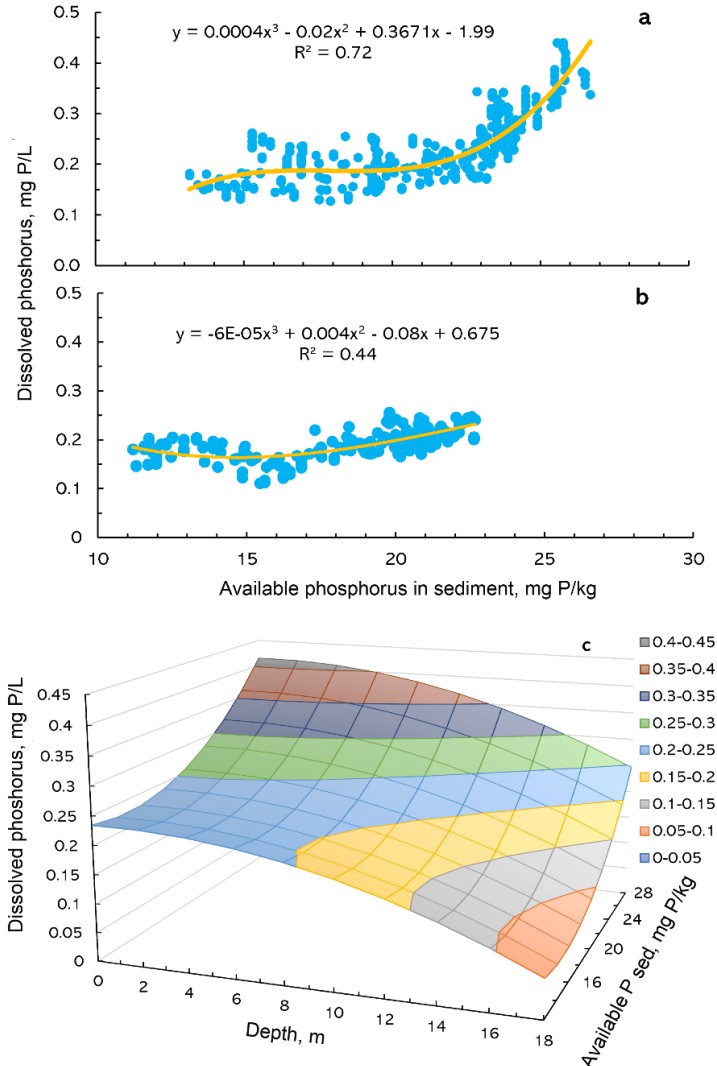

**Figure 6.** Phosphorus concentration of Lake Tana: (**a**) sediment available phosphorus concentration with dissolved phosphorus concentration for the northern part, (**b**) sediment available phosphorus concentration with dissolved phosphorus concentration for the southern part, and (**c**) 3D surface representation for prediction dissolved phosphorus from available phosphorus and lake depth.

The strong relationship between the dissolved and available phosphorus and depth is the directly related to the lake characteristics (Table 2) since all morphometric parameters indicate that the lake is mixed: the wave base depth ($D_{wb}$), defined as the maximum depth at which a water wave's passage causes significant water motion, is 14.8 m and thus extends up to Lake Tana's maximum depth of 14.8 m [19,45]. In addition, the relative depth, Drel, is 2.4%, which, according to Table 2, is less than the critical value of 5% for the lake to be stratified [1,46]. In addition, the dynamic ratio is 5.7 km/m, indicating that sediment resuspension dominates (Table 2). Finally, the energy topographic factor indicates that the sediment in 99% of the lake is suspended (Table 2) [13,47].

The correlation coefficient between depth and dissolved and available phosphorus was greater in the northern half of the lake than the southern part in March near the end of the dry monsoon phase, when the discharge of the Megech, Gumara and Rib (entering the lake in the northern part; Figure 1) are very small. Rivers have very little flow from January to the beginning of the rain phase in May

or June. At the same time, the water depth decreases. As a result, resuspension due to wind/wave action determines the sediment dynamics [19] and an equilibrium between the dissolved phosphorus and available phosphorus can be established. In the southern part, the Gilgel Abay that contributes nearly 60% of the flow [25] and the water (together with dissolved phosphors and sediment) that is leaving through the outlets to the Tana Beles Tunnel and the Blue Nile impedes the establishment of an equilibrium between dissolved phosphorus and the available phosphorus (Figure 1). Hence, the correlation in the southern half was weaker and the concentrations were lower. Thus, although the literature generally reports that the interaction of phosphorus in water and sediment is complex [48–50], we find a distinct relationship for a shallow tropical highland lake where an equilibrium exists between the sediment and water column [51].

### 3.5. Implication for the Spread of Water Hyacinths

Phosphorus is the limiting nutrient for the growth of water hyacinths [52]. Thus, areas with the greatest dissolved phosphorus concentration will have the greatest potential for the growth of water hyacinths [36]. As shown here, dissolved phosphorus concentrations are high where the available phosphorus is high and the lake depth is the shallowest (this is in the northeast, where the lake is shallow and the available phosphorus is high; Figures 3b and 4). Other parameters such as a pH above neutral, elevated tropical temperatures and relatively low dissolved oxygen is all enhance the growth of water hyacinths [36]. In addition, the wind is mainly in the northeast direction, preventing the water hyacinths from floating away. Due to this reason, the spatial extent of the water hyacinth is limited to the northeast of Lake Tana [42]. Due to the large pool of phosphorus in the sediment, controlling the spread of water hyacinths could be a lengthy and challenging process in Lake Tana [53]. Since phosphorus cycles from the bottom sediment to the water hyacinths and the biomass accumulates it, harvesting the plants will aid in reducing the bottom sediment concentration.

### 3.6. Comparison to Other Tropical Lakes

In comparing the morphometry of Lake Tana, most of the African great lakes are deep and stratified. For example, the wave base depth in Lake Malawi (with an average depth of 292 m and area of 28,800 km$^2$) and Lake Tanganyika (with an average depth of 570 m and area of 32,600 km$^2$) [5] is about 41 m, which represents 14% and 7% of the mean depth, respectively. The dynamic ratio is 0.6 less than 3.8 for Lake Malawi and 0.3 for Lake Tanganyika, indicating that phosphorus release from the bottom sediments is unlikely to occur [12].

In the largest African lake, Lake Victoria, with an area of 68,800 km$^2$, an average depth of 40 m, maximum depth of 82 m and a volume of 3200 km$^3$ [54], the wave base depth is about 42.3 m, which is approximately the same depth as the mean depth [55]. The dynamic ratio is 6.6. Thus, the stratification of Lake Victoria could be limited to the deep area that exceeds the mean depth and most of the bottom sediments are resuspended into the water column. Hence, the internal loading of phosphorus in areas shallower than the wave base depth is likely to occur [10]. Therefore, the invasion of water hyacinths in Lake Victoria is the result of the shallow morphology and internal phosphorus loading.

Lake Abaya in Ethiopia, with an area of 1140 km$^2$ and a mean depth of 8.6 m, is mixed similar to Lake Tana, with a dynamic ratio (DR) = 3.9 [56], and the primary production is low due to the high sediment load [57]. Finally, in Lakes Hayq and Hardibo, with average depths of 32.7 m and 25.5 m, respectively [58,59], the wave base depths are 8.4 m and 7.2 and the lakes are therefore stratified and water quality is relatively good compared with the shallow lakes [60].

### 4. Conclusions

The objective of the study was to determine the effect of lake morphometry on the sediment and phosphorus dynamics of Lake Tana. Morphometric characteristics were derived from a bathymetric survey conducted in late 2017. Both 2D and 3D bathymetric maps were generated from the bathymetric survey. The total storage capacity of the lake was 29.6 km$^3$. The lake has a nearly cylindrical shape with

an average depth of 9.7 m and a maximum depth of 14.8 m. Lake Tana is characterized as a shallow lake that continuously mixes. Consequently, the bottom sediments are constantly resuspended and the available phosphorus in the bottom sediment is in equilibrium with the dissolved phosphorus in the water columns. The greatest dissolved phosphorus concentrations were found in the shallow lake area, with the greatest amount of available phosphorus found in the bottom sediments. This relationship was stronger in the northern half of the lake than the southern half because the Gilgel Abay, the largest river flowing into the lake, removed phosphorus from the lake with the water flowing out through the Blue Nile outlet. The location of the water hyacinths was in the shallowest part of the lake with the highest available phosphorus in the sediment and therefore dissolved concentration. This study implies that reducing phosphorus input in the lake will reduce the dissolved phosphorus concentrations, but the response will be slow because the lake water interacts with a large pool of phosphorus stored in the sediment of the lake.

**Supplementary Materials:** The following are available online at http://www.mdpi.com/2073-4441/12/8/2243/s1. Figure S1: Extracted shape files overlaid on the mosaiced sentinel-2 images of Lake Tana downloaded from https://earthexplorer.usgs.gov: (a) minimum lake level, (b) average lake level, (c) maximum lake level. Figure S2: Extracted sampling location for sediment available and dissolved phosphorus concentration of Lake Tana, Figure S3: A plot of observed and predicted depth measurements of Lake Tana: (a) exponential semi-variogram model of ordinary kriging interpolation method, (b) observed versus predicted depth plot for the 2017 bathymetric survey. Figure S4: Slope map of Lake Tana bottom topography derived from the bathymetric survey in 2017. Figure S5: Elevation–area and elevation–volume relationships of Lake Tana: (a) lake depth versus percentage of area curve, (b) lake depth versus the percentage of volume curve. Table S1: Sediment available phosphorus concentration collected from 60 bottom sediment samples and dissolved phosphorus concentration collected from 143 water samples of Lake Tana. Table S2: Descriptive statistics for sediment available and dissolved phosphorus concentration in bottom sediments and lake water. The data of the bathymetric survey are available at https://ecommons.cornell.edu/handle/1813/70155.

**Author Contributions:** M.G.K. has contributed to conceptualization, data collection, data analysis, writing the original draft manuscript and improving the manuscript based on the comments and suggestions of the coauthors. A.A.K. collected the dissolved phosphorus concentrations. S.A.T. aided in formulating the objectives and administered the project, supervision, review and editing. M.A.B. contributed to formulating the methods and reviewed and edited the draft. F.A.Z. provided comments and suggestions on the research and paper drafts. T.S.S. contributed to conceptualization, the overall content, structure of the paper, supervising and improved the English together with Peggy Stevens. All authors have read and agreed to the published version of the manuscript.

**Funding:** The research was funded by the EXCEED–SWINDON project "Excellence Center for Development Cooperation–Sustainable Water Management in developing countries", centered in the Technical University of Braunschweig, Germany (http://www.exceed-swindon.org) within the framework of the DAAD Programme and Blue Nile Water Institute of Bahir Dar University Ethiopia. Additional funding was provided by the Robert S. McNamara Fellowships Program of the World Bank.

**Acknowledgments:** We would like to acknowledge the assistance of Smachew Necho and Getahun Birra during the bathymetric survey. Finally, we would like to thank Peggy Stevens for improving the English of the manuscript and Steve Pacenka for archiving the data.

**Conflicts of Interest:** The authors declare no conflict of interest.

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
