# Peer review of "The Relationship of Lake Morphometry and Phosphorus Dynamics of a Tropical Highland Lake: Lake Tana, Ethiopia"

_water, doi:10.3390/w12082243_

Round 1

Reviewer 1 Report

The review of manuscript ID: water-874110

The reviewed manuscript concerns the morphometry and dynamics of phosphorus in a tropical lake Tana, located in Ethiopia. The aim of studies was to find relationship between lake morphometry and phosphorus in the water and sediment.

The subject is interesting and data are valuable since it gives information about less known matter of African lakes. Especially graphical part is well done and clear. One small comment - the bathymetric map of lake should be also present in the relative values, where shoreline is shown as 0 m and next isobaths as 1 m and so on. Bathymetric plan drawings directly from arcGis program are less friendly for interpretation by readers, which are less familiar with that kind of data.  Also the deeper point of the lake should be marked on the bathymetric map precisely.

Because of fact that I haven’t access to technical software programs I couldn’t open supplementary material files, and my review is based on main text only.

In my opinion manuscript needs major revisions, because some information included in manuscript sometimes is uncomplete or even mistaken.

  1. In the first line of abstract (line 16) Authors maintain that lakes hold most of world freshwater resources. It is mistaken statement because, the bigger freshwater source than lakes are: ice and glaciers, and also groundwater. It should be corrected .
  2. In the line 18 and 54 Authors give the info that the Tana Lake area is 3000 km2, but later in the Table 2 they present different value (3046 km2). In scientific publications it should be shown precisely eg. “Lake Tana (area 3046 km2, maximum depth 14.8 m)”.
  3. In the lines 47-48 Authors wrote that “wave base depth…separates the epilimnion …from hypolimnion”. It is partially not complete information. Stratification in lakes also depends on water temperature, and in stratified lakes epilimnion is separated from hypolimnion by metalimnion layer. Then results of calculated indicators such as wave based depth or relative depth should be supported by the real thermal profiles data. In lakes sometimes theoretically assessed data are little different than these really observed, and always the possibility of partial stratification existence. It is also possible, that partial water stratification could exist in that lake parts, where the water hyacinth appeared.

Also statement written in lines 259-260 should be corrected.

  1. Water temperature, oxygen conditions and water pH are crucial for phosphorus exchange in the water-sediment interface. Some information about these parameters should be included in text and discussed.
  2. In the description of study area (lines 75 -82) Authors mention, that water level in lake is strongly dependent on the work of hydropower plants. In my opinion the potential influence of these installations on functioning of studied lake should be discussed in the Discussion section, especially in the aspect of P internal loading possibilities.
  3. The shape of lake bowl could be easily characterized by depth index (Dmv/Dmax) value too.
  4. Also % of water change per year could be included in the table. This index is used in many lakes assessment systems (eg. lake susceptibility to degradation etc.).
  5. In the Table 2 the maximum depth and maximum length values should be given. Also in the line with relative depth the symbol “<” should be changed into “>”.
  6. The separated lake map with water hyacinth area also could be included in manuscript. Authors maintain in line 193 that it is shown in Figures 3 and 4 c. But Fig. 4 show completely different data, and Fig.3 also do not show it precisely.
  7. Water hyacinth biomass can act as a trap for nutrients and this aspect in my opinion could be mentioned in Discussion section.
  8. Authors should precisely define units used in text– mg P/l or mg PO43-/l, because it is ambiguous. For sediment P – mg P/kg of sediment wet weight of dry weight ?
  9. 5 a- in the legend it should be “Available” instead of “Availabe”

With kind regards,

Reviewer

Author Response

Thank you for the comments. Please see the PDF for the response and the marked-up manuscript

Tammo and Mebrahtom

Reviewer 2 Report

This articles is very interesting but you may improve the text? see comments in the manuscript.

Author Response

Dear Reviewer

Thank you for your comment. Our response and the marked-up manuscript are attached

Regards

Tammo and Mebrahtom

Reviewer 3 Report

In the submission, the authors provide a research study regarding assessment of Lake Tana’s morphometric parameters and its implication on water quality.

The work is useful for the scientific community due to the fact that lakes are significant ecosystems whose function is influenced by the water quality affected by natural and anthropogenic factors.

The strong issues of the submission are:

  1. The title reflects the content of the paper.
  2. The keywords are suitable so the article can be found in the current registers or indexes.
  3. The abstract is informative and completely self-explanatory. The abstract clearly identifies the chemical systems investigated, experimental approaches used, and summarize the results.
  4. The introductory part presents an overview on current research on the issue investigated. The literature is sufficiently evaluated.
  5. The structure of the article is according to the structure of a research article.
  6. The presentation reflects the present state of knowledge.
  7. The authors explain how the experiments were performed. There is sufficient information present for other researchers to replicate the research. The article identifies the procedures followed.
  8. The methods and protocols are described in sufficient detail in order to allow another researcher to reproduce the results.
  9. In the Conclusions part the authors mentioned the major and specific conclusions of their research study. All the conclusions are justified and supported by the experimental results.
  10. The authors related their findings to other researcher’s results.
  11. The text is presented in a manner that scientists in other disciplines will understand.
  12. The abbreviations and nomenclature are used according to the applicable international standards and rules.
  13. The figures and tables are numbered sequentially, and they are clearly labeled and positioned close to the relevant text. Titles of figures and tables are brief and informative. All the figures and tables included are referred.
  14. The size of the article is appropriate to the content.
  15. The references are accurate and relevant for the subject of the paper.
  16. The work fits into the wider literature regarding environmental pollution and remediation studies.
  17. The authors acknowledge for the financial and technical support.

However, there are some issues that have to be solved by the authors.

  1. The measuring units should be noted in agreement with the valid standards. I suggest to replace mg/l with mg/L, mg P/l with mg P/L.
  2. The authors should check the paper so that the indices of the units of measurement are written correctly (km2 should be replaced by km2).
  3. The reference part should be checked by the authors to be written according to the journal’s rules (the reference number 52 should be rewritten).

In my opinion, the paper analyzed addresses some important issues regarding the effect of lake morphometry on sediment and phosphorus dynamics.

The topic of this study is included in the important research fields of chemical engineering, environmental protection and environmental engineering, being relevant for environmental engineers but also for environmental protection authorities.

This paper can be published in the Water after minor changes.

Author Response

Dear Reviewer

Thank you for your positive evaluation of the manuscript.  In the attached PDF we have responded to your comments. The marked-up manuscript is appended to the response

Thank you again

Tammo and Mebrahtom

Reviewer 4 Report

Comments on:  The relationship of Lake morphometry and phosphorus dynamics of a tropical lake: Lake Tan Ethiopia

The manuscript is well organized  and deals with an interesting aspect. There are only minor corrections necessary.

L71: verb is missing

L160-162: some abbreviations are already explained in the table above

Table 2: Relative depth, mean slope and energy topography factor: from their formulas I guess that fraction is the correct term; a fraction does not have a dimension. In case you mean percentage  you have to multiply these formulas by 100.  

The dimension of area and volume are km2 and km3, respectively.

L203: Where is the Chara Chara Weir? At Tana Beles or Abay R.?

L253+254: better: …sediment-available…

L261-264: There should be a blank between figures and %.

L266-267: awkward sentence

L280-282. Omit repetitions. Therefore: …concentrations are high, where the lake is shallowest. This is in the northeast of the Lake.  In my view the spread of the water hyacinths is the wind blowing towards the northwest (as you say). Thus, the coincidence of distribution of the hyacinths and in areas of shallow lake could simply be a physical effect and not necessarily a chemical one. As far as I know these hyacinths are not rooted in the sediment. So, how can you prove the relation between phosphorus and the presence of the hyacinths?

Author Response

Dear Reviewer

Thank you so much for your review.  Our response to your comments is in the attached PDF together with the marked-up mauscript

Regards

Tammo and Mebrahtom

Round 2

Reviewer 1 Report

Second review of manuscript ID water-874110

Authors substantially improved manuscript according to reviewer's suggestions.

I have only one small comment - in the line 135 the word "wight" should be changed into "weight".

In my opinion manuscript can be publish in Water in the present form.

With kind regards

Reviewer